# Identifying Frequently Used NANDA-I Nursing Diagnoses, NOC Outcomes, NIC Interventions, and NNN Linkages for Nursing Home Residents in Korea

**DOI:** 10.3390/ijerph182111505

**Published:** 2021-11-01

**Authors:** Juh Hyun Shin, Gui Yun Choi, Jiyeon Lee

**Affiliations:** 1College of Nursing, Ewha Womans University, Seoul 03760, Korea; 2Department of Nursing, Ulsan College, Ulsan 44022, Korea; gychoi@uc.ac.kr; 3College of Nursing, Catholic University of Pusan, Busan 46252, Korea; jylee@cup.ac.kr

**Keywords:** nursing homes, standardized nursing languages, NANDA-I, NOC, NIC

## Abstract

This study aimed to identify the terminologies of NANDA-I, NOC, NIC, and NNN linkages that have been used for nursing home (NH) residents. This study used a retrospective descriptive design. Data accrued from 57 registered nurses (RNs) in 25 Korean NHs. The RNs randomly selected one resident and assessed for applied NANDA-I, NOC, and NIC from the previous 7 days by reviewing nursing charts and records. Finally, the data of 57 residents in 25 NHs were collected. Results: We identified seven NNN linkages: risk for falls–fall prevention behavior–fall prevention; self-care deficit: bathing/hygiene–self-care: activities of daily living (ADL)–self-care assistance: bathing/hygiene; impaired memory–memory–cognitive stimulation; chronic confusion–neurological status: consciousness–medication management; chronic confusion–memory–medication management; impaired walking–mobility–exercise promotion: strength training; and impaired walking–ambulation–exercise promotion: strength training. The identified core NANDA-I, NOC, NIC, and NNN linkages for NH residents from this study provide a scope of practice of RNs working in NHs.

## 1. Introduction

The nursing process is an independent clinical judgment in which a registered nurse (RN) assesses an individual’s health to diagnose actual or potential health problems [1]. The nursing process is a systematic method that allows RNs to effectively care for patients using problem solving and critical thinking [2]. The aim of the nursing process is identifying, diagnosing, and treating actual or potential human responses to disease [3]. The nursing process helps nurses make professional judgments in terms of clinical and problem-solving methods and nursing management [2,3]. A standardized nursing-languages system scientifically and efficiently applies the nursing process to individuals [3]. The standardized nursing-languages system objectively expresses and conceptualizes the phenomenon of nursing and clarifies the nursing process of individuals by using individuals’ problems in a common technical term [3,4].

The NANDA-I, NOC, and NIC are the most common standardized nursing languages systems [5]. Using NANDA-I provides the basis for selecting nursing interventions to achieve outcomes for which the RN is accountable and gives RNs a standardized language to articulate problems they encounter daily [6]. NANDA-I prioritizes the most urgent needs of the patient [7]. The NOC shows detailed outcome measurements to RNs and supplies the intermediary outcomes, which helps accomplish long-term outcomes [6,8]. The NIC is an intervention from NOC. Using NIC enables RNs to focus nursing behaviors, which helps solve the nursing problem [6,8]. Creating and using NANDA-I, NOC, NIC, and NNN linkages enables holistic nursing care appropriate for an individual’s illness/health condition [9,10]. NNN linkages assist the RN in making decisions about the outcome and intervention of care plans [11]. Through using NNN linkages, RNs continually evaluate the situation and adjust NNN to fit the unique and diverse needs of each patient [6,11]. It is easier for RNs to apply nursing languages to practice if the nursing home (NH) staff have brief and clear nursing-language linkages developed for their specific setting; the desired and suggested outcomes guide RNs toward what they should do for residents [6]. The continuity of care through standardized languages contributes to better outcomes not only for patients but also for nursing staff [4]. Using consistent standardized languages improves the quality of patient outcomes and advances nursing knowledge and practice [4,9,12].

Although standardized nursing languages systems apply in many settings (due to their importance) the systems previously focused only on individuals in acute settings [9], and few studies have described the use of language systems in NHs. NH residents are a very vulnerable population. Most residents have at least one chronic or geriatric illness, and most require constant professional nursing care in long-term-care settings [13]. Implementing standardized terminology systems in NHs improves monitoring quality, payment for residents, outcome research, and decision support [14]. Documents in NHs should include nursing-oriented standardized languages because nurses play a more independent and critical role as case managers in NHs, compared with acute settings like hospitals. However, using a standardized nursing language in an NH is very rare because NHs in Korea lack a uniform and standardized nursing-care record system. Worldwide, Korean NHs do not have a foundation in any standardized terminology system.

It is easier and more efficient for RNs to apply nursing languages to their practice if the NH staff have brief and clear nursing-language linkages developed only for their setting; the desired and suggested outcomes guide RNs toward how they should care for the residents [9]. The frequently used nursing diagnoses and interventions with desired outcomes, developed in several studies, can guide newly graduated or hired NH RNs [15]. Nursing language linkages in NHs may help RNs define the appropriate nursing interventions by nursing diagnosis and establish nursing outcomes for the elder population [14]. However, data on nursing diagnosis, intervention, outcomes, and the linkages in NHs are limited, despite the importance of a standardized nursing-languages system. This study identified the frequently used terminologies of NANDA-I, NOC, NIC, and NNN linkages that have been used for NH residents.

## 2. Methods

### 2.1. Design of the Study

This study used a retrospective descriptive design. The conceptual framework of this study is the structure–process–outcome (SPO) model developed by Unruh and Wan [16] for evaluating the quality of long-term care facilities. NANDA is classified in the structure, NIC is equivalent to the process, and NOC belongs to the outcome.

### 2.2. Setting and Sample

We randomly selected 140 NHs (20% of total NHs) using the random function of the Excel program and contacted NHs with RNs listed on the Korean Long-Term Care homepage operated by the Korean National Health Insurance Corporation [17]. The Korean RN-staffing regulations in NHs allow certified nursing assistants (CNAs) to replace RNs; only 700 NHs (about 21.9%) of 3200 NHs operating in 2019 employed RNs. We explained the purpose and process of this study to NH administrators and requested participation in this study by email or phone (due to COVID-19) using the e-mail address and phone number posted on the Korean Long-Term Care homepage. Administrators of 25 NHs voluntarily agreed to participate in this study. Among the agreed NHs, RNs who wished to voluntarily participate in this study were included. A total of 57 RNs from 25 NHs participated in this study. Each RN randomly selected one resident using the random-selection method. The criteria for including residents was those aged 65 or older who have been admitted for more than 6 months. We recruited RNs with more than 5 years of experience in NHs to participate. RNs who do not provide direct care to residents (i.e., educational RNs, administrative RNs) were excluded from this study. We provided financial incentives of KRW 35 for their survey completion.

### 2.3. Instruments

#### 2.3.1. General Characteristics

Resident information on sex, age, resident case-mix using a Korean long-term care grading system, underlying medical conditions, and length of stay were collected through admission records.

#### 2.3.2. NANDA-I

NANDA-I involves a clinical judgment about individual, family, or community responses to actual and potential health problems/life processes and offers the basis for choosing nursing interventions to accomplish nursing outcomes [18]. The nursing diagnoses in this study originated from 221 NANDA-I nursing diagnoses translated into Korean by Choi [19]. The diagnoses used in this study were extracted from Ackley’s book [20]. We selected 45 nursing diagnoses, building on nine previous studies of nursing diagnosis for residents of NHs. Most NANDA-I nursing diagnoses in NHs are as follows: Self-Care Deficit: Bathing/Hygiene, Self-Care Deficit: Dressing/Grooming, Impaired Physical Mobility, Altered Thought Process, and Potential for Injury [21,22]. The eight never-used NANDA-I nursing diagnoses, which do not relate to the elderly at all, were excluded from this study and are as follows: Effective Breastfeeding, Ineffective Breastfeeding, Ineffective Infant Feeding Pattern, Interrupted Breastfeeding, Potential Altered Parenting, Rape-Trauma Syndrome: Compound Reaction, and Rape-Trauma Syndrome: Silent Reaction. In addition, the 201 relative factors and 128 defining characteristics were included in each of the 45 diagnoses.

#### 2.3.3. NOC

The NOC is an all-inclusive, standardized classification of patient/client outcomes to evaluate the effects of nursing interventions [23]. The nursing outcomes in this study originated from 385 nursing outcomes developed by the University of Iowa research team translated into Korean by Choi [19]. The outcomes used in this study were extracted from Ackley’s book [20]. We selected 79 nursing outcomes to construct the survey, building on the previous nine studies of nursing outcomes for residents in NHs. Most NOCs in the previous NH study are as follows: bowel elimination, urinary elimination, memory, health-promoting behavior, and neurological status: consciousness [9].

#### 2.3.4. NIC

The NIC is defined as any direct-care treatment an RN performs on behalf of a client. The nursing interventions in this study originated from 211 NIC developed by the University of Iowa research team translated into Korean by Choi [19]. The interventions used in this study were extracted from Ackley’s book [20]. We selected 82 nursing interventions to construct the survey, building on the previous nine studies of nursing interventions for residents in NHs. In previous studies, the core NIC interventions included active listening, behavior management, caregiver support, communication enhancement, and confusion management [24]. Two nursing professors with clinical experience in NHs and NH investigation selected and screened nursing diagnoses, outcomes, and interventions. Among the two professors, the first researcher (a professor who has clinical experience in NHs and whose main research field is NHs) selected nursing diagnoses, outcomes, and interventions from the previous nine studies of NHs. The second researcher (a professor focusing on geriatric care, and a policy expert in long-term care) screened the selected nursing diagnoses, outcomes, and interventions for NH applicability, and analyzed the possibility of further applicable nursing diagnoses, outcomes, and interventions. Finally, we used 45 nursing diagnoses, 79 nursing outcomes, and 82 nursing interventions in the study’s questionnaire.

### 2.4. Method of Data Collection

We collected data from January to February 2020 after the institutional review board of a university in Korea approved the study (Approval No. XX-202001-0002-01). We visited the NHs (or sent information by mail to NHs that refused visits due to COVID-19) and explained the purpose and procedure of the study to the RNs and obtained consent forms. We explained the definition, purpose, and brief history of nursing diagnoses, outcomes, and interventions using a booklet and how we developed questionnaires before providing the survey so RNs had no difficulty filling out the questionnaire. RNs selected one resident (one resident for one RN) using a random sampling method by using excel program and referred to the resident’s RN records from the previous 7 days to complete the questionnaire. RNs were asked to mark “Applied for nursing diagnosis, intervention, and outcome applied for 7 days and “Not Applied if it was not applied. RNs provided information on residents’ general characteristics from the admission records. For the interrater reliability of the data, two RNs independently checked nursing diagnoses, outcomes, and interventions on the questionnaire. If they chose different nursing diagnoses, outcomes, and interventions, they were unified into one after the two RNs discussed their findings. This process secured the reliability between the observers.

### 2.5. Data Analysis

We analyzed data using the Statistical Package for the Social Sciences for Windows Version 25.0. (SPSS Inc., Chicago, IL, USA). We analyzed the general characteristics of NH residents using numbers and percentages. We analyzed the frequency of nursing diagnoses, interventions, and outcomes among the NH residents by numbers and percentages. Researchers linked NNN standardized languages. We integrated NNN association linkages by combining NIC (year) and NOC (year) taxonomies, in terms of the analogous domains and classes. We started with the NANDA-1 as an outline followed by the NIC and NOC. The whole process is parallel to the nursing process [25].

## 3. Results

### 3.1. General Characteristics of Residents in NHs

Table 1 provides a descriptive analysis of the 57 residents in 21 NHs. Residents were predominantly female (42; 74%), their mean age was 85.19 years (SD = 6.38), and the average length of stay was 3 years and 1 month (SD = 40.36 months). Of residents, 40.3% were third grade long-term-care beneficiaries (residents who are partially dependent for activities of daily living (ADLs)). Residents diagnosed with dementia totaled 94.7%; 68.4% were diagnosed with high blood pressure, 31.4% with diabetes, and 29.8% with a neurological disease.

### 3.2. Most Frequently-Used NANDA-I Nursing Diagnoses, NOC Outcomes, and NIC Interventions in NHs

Table 2 shows the top 30 NANDA-I nursing diagnoses, relative factors, and defining characteristics NHs most frequently used. The most frequent nursing diagnosis was risk for falls (applied to 49 of 57 residents), and the most relative factor was residents over 65 years of age. The second most frequent nursing diagnosis was self-care deficit: bathing/hygiene; the highest relative factor for self-care deficit: bathing/hygiene (applied to 44 residents) indicated cognitive impairment, and the defining characteristic was the inability to wash oneself. The third most frequent nursing diagnosis was impaired memory; the highest relative factor for impaired memory (applied to 41 residents) was a neurological problem, and the defining characteristic was that no action was taken. The fourth most frequent nursing diagnosis was chronic confusion (applied to 41 residents); the highest relative factor for chronic confusion was Alzheimer’s disease, and the defining characteristic was short-term memory loss. The fifth most frequent nursing diagnosis was impaired walking (applied to 39 residents); the highest relative factor for impaired walking was insufficient muscle strength, and the defining characteristic was the inability to walk a required distance.

Table 3 describes the top 30 NOC outcomes NHs most frequently used. The most frequently applied NOC outcome was vital signs. This outcome was applied to most residents (56 out of 57). The second and third most frequently applied NOC outcomes were comfort status and nutritional status: nutrient intake, separately applied to 51 residents. The fourth and fifth most frequently applied NOC outcomes were oral hygiene and fall-prevention behavior, separately applied to 50 residents.

Table 3 also shows the top 30 NIC interventions most frequently used in NHs. The most frequently applied NIC intervention was medication management. This intervention was applied to most residents (56 out of 57). The second most frequently applied NIC intervention was vital-signs monitoring, applied to 55 residents. The third most frequently applied NIC intervention was environment management: comfort. This intervention was applied to 54 residents. The fourth most frequently applied NIC intervention was fall prevention, applied to 53 residents. Surveillance: safety was the fifth most frequently applied nursing intervention, applied to 52 residents.

### 3.3. Linkages of Nursing Diagnoses, Outcomes, and Interventions in NHs

Table 4 shows the top seven NNN linkages for residents in NHs. Two thousand nine hundred fifty-eight different NNN linkages were used for residents in NHs. The seven most commonly used NNN linkages are as follows: risk for falls–fall prevention behavior–fall prevention; self-care deficit: bathing/hygiene–self-care: ADL–self-care assistance: bathing/hygiene; impaired memory–memory–cognitive stimulation; chronic confusion–neurological status: consciousness–medication management; chronic confusion–memory–medication management; impaired walking–mobility–exercise promotion: strength training; and impaired walking–ambulation–exercise promotion: strength training.

For risk for falls, we selected one NOC outcome—fall prevention behavior—and the most frequently used NIC interventions included fall prevention and surveillance: safety. For the self-care deficit: bathing/hygiene, we selected one NOC outcome—self-care: ADL, and the most frequently used NIC intervention was self-care assistance: bathing/hygiene. For impaired memory, we selected one NOC outcome—memory—and the most frequently used interventions included active listening and memory training. For chronic confusion, the most frequently used NOC outcomes included neurological status: consciousness, and memory. Medication management, vital-signs monitoring, surveillance: safety, environment management: safety, and dementia management were the most frequently selected interventions for the nursing outcome of neurological status: consciousness. Medication management, cognitive stimulation, and memory training were the most frequently selected interventions for the nursing outcome of memory. For impaired walking, we selected two NOC outcomes—mobility and ambulation—and the most frequently used corresponding interventions included exercise promotion: strength training, and exercise therapy: joint mobility.

## 4. Discussion

This study is quite meaningful because a specialized NNN linkage was developed by linking NANDA-I nursing diagnoses, NOC outcomes, and NIC interventions with high frequency only for NH residents. We identified the most frequently used NANDA-I nursing diagnoses, NOC outcomes, and NIC interventions and the links between nursing languages in Korean NHs.

The most prevalent NANDA-I nursing diagnoses included risk for falls (86.0%) and impaired walking (78.0%), which led to a decreased ability to ambulate and limited the performance of ADLs. Consistent with previous data [14], the professional management of falls is a top priority in NHs because NH residents are vulnerable to falls. The prevalence of falls in global NHs is 24.9%–37.3% [26,27] and approximately 25.2% of Korean NH residents have fallen [28]. NH residents are vulnerable to falls due to old age. Aging causes a decrease in muscle strength and tendon elasticity [29]. We found insufficient muscle strength is the factor most relative to impaired walking. Increased risk for falls and reduced ability to walk reduces opportunities for individuals to participate in functional physical activities necessary to perform self-care, leading to a loss of muscular strength and triggering a vicious circle of inactivity and weakness [30]. Beyond aging factors, a higher proportion of RNs in NHs and greater RN hours per resident day align with a decreased risk of fall injuries [28,31] because RNs play a key role in fall prevention. In NHs, RNs assess fall-risk factors (e.g., physical restraints or urination problems), remove fall-risk factors, create a fall-prevention environment (e.g., bedside arrangement or raising the bed railing), and distribute the role of nursing assistants to prevent falls [32]. Therefore, appropriate levels of RN staffing must be secured for fall prevention.

Bathing is the ADL with the highest disability incidence in older people. This diagnosis was used at a high frequency in preceding studies [11]. This diagnosis relates to the sequential order of ADL decline. First, elders lose their ability to bathe independently; second, they lose their mobility independence; and third, their ability to eat independently [11]. Bathing aids body hygiene and skin integrity, which are vital to the prevention of disease. Furthermore, NH residents’ daily activities, such as bathing, positively impact their quality of life [33]. Despite these positive effects, NH residents do not take a bath for reasons including disability or depression [34]. RNs in NHs are responsible for assessing which difficulties residents have with bathing, planning a resident’s bathing activities, encouraging residents to take a bath, and assigning nursing staff to assist residents in taking a bath [34]. Greater RN hours per resident day associate with increased resident bathing [35].

Dementia management is another priority among nursing care, as it causes irreversible changes in cerebral function, memory disorders, roles, and daily activities [36]. Accordingly, impaired memory and chronic confusion are prevalent nursing diagnoses in this stimulation and were applied at high frequency for residents as an intervention for impaired memory. RNs can identify cognitive deficits through a mental status examination, and plan and provide cognitive-improvement programs for residents [37]. For example, RNs can provide one cognitive-stimulation intervention (enriched living environments that contain a wide array of personal memorable and memory-stimulating cues) to potentially support NH residents’ cognitive functioning [38,39]. An enriched environment provides items to simulate residents in meaningful ways. An enriched environment includes making residents’ rooms more homelike, preserving a sense of self, stimulating autobiographical memory, and helping residents recall current and past relationships [39]. In this study, chronic confusion in residents was applied with high frequency and the highest frequency of intervention in chronic confusion was medication management. A higher ratio of RNs to CNAs and a higher number of RN hours per resident day equate to a lower number of patients with chronic confusion [40]. Nursing staff have difficulty assessing chronic confusion [41]. Due to this characteristic, interventions for chronic confusion are not adequately conducted [41]. However, RNs receive more extensive clinical training and education on problem-solving skills compared to other nursing staff. With this training and education, RNs have tacit and articulable knowledge to better assess chronic confusion in NH residents [40]. Therefore, appropriate levels of RN staffing should be performed for chronic confusion management.

Walking ability is fundamental to independence [42]. Impaired walking is a strong indicator for admitting a person to a NH [43]. Impaired walking causes disabilities, falls, and fractures. Therefore, strength training is necessary for NH residents to prevent further disabilities and falls [44].

Using a standardized nursing language in NHs has numerous benefits. First, a standardized nursing language enhances communication among nursing staff, healthcare professionals, and administrators of institutions [45]. Effective communication is necessary when healthcare practitioners transfer work in a NH, or when residents move to another NH, hospital, or emergency room. A standardized nursing language improves communication and enhances resident care by facilitating seamless communication between healthcare practitioners. Second, a standardized nursing language increases the visibility of nursing interventions [46]. The public do not know exactly what care RNs perform, thereby making it difficult to understand RNs’ professional nursing practices. RNs do not use a standardized nursing language and have a long tradition of overreliance on communicating information through word-of-mouth [46,47]. A standardized nursing language helps highlight the contribution of NH RNs to resident outcomes, thus enhancing nursing visibility. A standardized nursing language enables RNs to describe the unique role of the RN. Third, a standardized nursing language enables RNs to provide optimal and exact nursing interventions to residents. When a specific nursing diagnosis is given to a resident, RNs can quickly understand the standards of care through a standardized nursing language. Therefore, providing a standardized nursing language is needed when educating nursing students [15]. A standardized nursing language can improve residents’ quality of care by promoting communication among the nursing staff and thereby standardize nursing practices. Considering these benefits, NHs must use the NNN developed in this study.

A limitation of this study includes that a 1-week review of the RN record was limiting due to the 4-month grant period, which was not a long enough for data collection. Additionally, the number of subjects (RNs) and residents selected by the RNs included in this study is small to reflect whole NH residents. In future studies, it is necessary to secure a sufficient data-collection period and increase the number of nursing records of residents reviewed per RN. Third, the understanding of nursing diagnoses, outcomes, and interventions is different for each RN due to differences in competency, including education level and understanding of nursing diagnoses, outcomes, and interventions. Therefore, we only included RNs with more than 5 years of clinical experience in NHs for this study. Despite these limitations, this study was very valuable because it is one of the first trials conducted with RNs in NHs and can be identified by most frequently applying which nursing diagnoses, outcomes, and interventions RNs perform. Based on this NNN linkage, future studies must be conducted on whether linkages are valid for various NHs throughout the country.

## 5. Conclusions and Practice Implications

We identified nursing diagnoses, outcomes, interventions, and NNN linkages applied in NHs. Standardized nursing languages for nursing diagnoses, outcomes, and interventions in NHs provide a common language for communication, evidence of the unique and professional roles of RNs, and educational material for newly graduated or hired NH RNs. A standardized nursing language enables RNs to provide optimal and exact nursing interventions to residents. In conclusion, the identified core NANDA-I, NOC, NIC, and NNN linkages for NH residents from this study provide a scope of practice of RNs working in NHs. The linkage can be applied to RNs’ independent activities with nursing languages to solve the potential or possible risk problems. Therefore, preparing and supporting RNs to plan, perform individualized nursing on identified problems, and evaluate resident responses on the basis of a standardized nursing language is essential. These efforts lead to improved quality of care for NH residents.

## Figures and Tables

**Table 1 ijerph-18-11505-t001:** Residents’ General Characteristics (*n* = 57).

Variables	Categories	*n*	%	M ± SD	Range
Age (years)				85.19 ± 6.38	67–98
Sex	Male	15	26.0		
	Female	42	74.0		
Long-term care grade	1 ^a^	8	14.0		
	2 ^b^	12	21.1		
	3 ^c^	23	40.3		
	4 ^d^	13	22.8		
	5 ^e^	1	1.8		
Length of stay (month)			37.26 ± 40.36	1–245	
History	Hypertension	39	68.4		
	Diabetes	18	31.6		
	Thyroid	5	8.8		
	Cardiac disease	8	14.0		
	Respiratory disease	4	7.0		
	Neurological disease	17	29.8		
	Dementia	54	94.7		
	Arthritis	9	15.8		
	Osteoporosis	11	19.3		
	Parkinson’s disease	5	8.8		

Note. ^a^ = a person who is in complete need of other people’s help in everyday life; ^b^ = a person who needs the help of others in much of everyday life; ^c^ = a person who partially needs the help of others in everyday life; ^d^ = a person who needs help in some part of everyday life; ^e^ = dementia.

**Table 2 ijerph-18-11505-t002:** Frequency of Top 30 NANDA-I nursing diagnoses, related factors, and defining characteristics.

Rank	NANDA-I Nursing Diagnoses (*n* = 1007)	*n*	%	Related Factors	*n*	%	Defining Characteristics	*n*	%
1	Risk for falls	49	86.0	Age > 65	29	59.1	—		
2	Self-care deficit: bathing/hygiene	44	77.2	Cognitive impairment	23	52.2	Inability to wash oneself	24	54.5
3	Impaired memory	41	71.9	Neurological problem	21	51.2	Unknown action taken	21	51.2
4	Chronic confusion	41	71.9	Alzheimer’s disease	22	53.7	Short-term memory loss	21	51.2
5	Impaired walking	39	68.4	Insufficient muscle strength	21	53.8	Unable to walk the required distance	19	48.7
6	Constipation	35	61.4	Lack of physical activity	21	60.0	Non-formal symptoms seen in the elderly	13	37.1
7	Risk for infection	34	59.6	Chronic disease	20	58.8	—		
8	Ineffective health maintenance	34	59.6	Cognitive impairment	17	50.0	Personal damage support system	12	35.3
9	Impaired physical mobility	34	59.6	Reduced muscle control	16	47.1	Restricted ROM	13	38.2
10	Powerlessness	33	57.9	Powerless lifestyle	17	51.5	Depression	13	39.4
11	Activity intolerance	32	56.1	General weakness	17	53.1	Exercise delay	10	31.2
12	Risk for constipation	31	54.4	Insufficient physical activity	16	51.6	—		
13	Sedentary lifestyle	29	50.9	Lack of interest	12	41.4	Physical weakness	13	44.8
14	Impaired urinary elimination	27	47.4	Various factors	14	51.9	Urinary incontinence	11	40.7
15	Anxiety	27	47.4	Threat or change	11	40.7	Sleep disorder	7	25.9
16	Impaired verbal communication	27	47.4	Perception disorder	16	59.3	Difficult to understand communication	13	48.1
17	Functional urinary incontinence	26	45.6	Cognitive impairment	14	53.8	Urination before arriving in the bathroom	7	26.9
18	Risk for peripheral neurovascular dysfunction	26	45.6	Not moving	12	46.2	—		
19	Risk for impaired skin integrity	26	45.6	Stool	12	46.2	—		
20	Chronic pain	24	42.1	Chronic physical disability	14	58.3	Pain appeal	10	41.7
21	Self-care deficit: feeding	24	42.1	Cognitive impairment	13	54.2	Unable to swallow food	10	41.7
22	Impaired bed mobility	22	38.6	-			Unable to change body position in bed	11	50.0
23	Disturbed sleep pattern	21	36.8	Sleep partnership	7	33.3	Normal sleep type changed	6	28.6
24	Acute pain	20	35.1	Damaging factor	2	10.0	Face expression	8	40.0
25	Impaired adjustment	20	35.1	Pessimistic	10	50.0	—		
26	Risk for loneliness	19	33.3	Physical isolation	6	33.3	—		
27	Stress urinary incontinence	17	29.8	Degenerative changes in pelvic muscle	10	58.8	Urination without the contraction of the urinary muscle	9	52.9
28	Bowel incontinence	17	29.8	Loss of anal sphincter control	9	52.9	Unrecognized desire to defecate	8	47.1
29	Risk for disuse syndrome	17	29.8	Unable to move one’s body	9	52.9	—		
30	Disturbed sensory perception: visual	16	28.1	Illusion	9	56.3	Loss of cognition	9	56.3

**Table 3 ijerph-18-11505-t003:** Frequency of Top 30 NOC outcomes and NIC interventions.

Rank	NOC Outcome (*n* = 2603)	*n*	%	NIC Intervention (*n* = 2955)	*n*	%
1	Vital Signs	56	98.2	Medication Management	56	98.2
2	Comfort status	51	89.5	Vital Signs Monitoring	55	96.5
3	Nutritional Status: Nutrient Intake	51	89.5	Environment Management: Comfort	54	94.7
4	Oral Hygiene	50	87.7	Fall Prevention	53	93.0
5	Fall Prevention Behavior	50	87.7	Surveillance: Safety	52	91.2
6	Communication: Expressive	49	86.0	Cognitive Stimulation	52	91.2
7	Personal Well-Being	45	78.9	Environment Management: Safety	52	91.2
8	Communication: Receptive	45	78.9	Teaching: Prescribed Medication	51	89.5
9	Psychosocial Adjustment: Life Change	44	77.2	Oral Health Maintenance	51	89.5
10	Personal Safety Behavior	43	75.4	Nutrition Management	51	89.5
11	Social Support	42	73.7	Self-care Assistance: Dressing/Grooming	51	89.5
12	Appetite	42	73.7	Bowel Management	50	87.7
13	Risk Control	42	73.7	Mood Management	50	87.7
14	Health Promoting Behavior	41	71.9	Emotional Support	50	87.7
15	Joint Movement	41	71.9	Infection Control	49	86.0
16	Balance	41	71.9	Exercise promotion: Strength Training	48	84.2
17	Neurological Status: Consciousness	40	70.2	Temperature Regulation	48	84.2
18	Self-Care: Eating	40	70.2	Body Mechanics Promotion	48	84.2
19	Skeletal Function	39	68.4	Exercise Therapy: Joint Mobility	47	82.5
20	Bowel Elimination	39	68.4	Self-care Assistance: Bathing/Hygiene	47	82.5
21	Anxiety Level	39	68.4	Dementia Management	47	82.5
22	Nutritional Status: Food & Fluid Intake	39	68.4	Support System Enhancement	46	80.7
23	Symptom Control	39	68.4	Skin Surveillance	45	78.9
24	Thermoregulation	39	68.4	Hope Instillation	45	78.9
25	Body Positioning: Self-Initiated	39	68.4	Coping Enhancement	45	78.9
26	Mobility	38	66.7	Nutritional Monitoring	44	77.2
27	Self-care: Activities of Daily Living (ADLs)	38	66.7	Activity Therapy	44	77.2
28	Tissue Integrity: Skin & Mucous Membranes	38	66.7	Infection Protection	44	77.2
29	Ambulation	38	66.7	Memory Training	43	75.4
30	Memory	37	64.0	Exercise therapy: Ambulation	43	75.4

**Table 4 ijerph-18-11505-t004:** 7 NNN linkages frequently used in NHs.

	NANADA-I Nursing Diagnoses	NOC Outcome	NIC Intervention
1	Risk for falls	Fall prevention behavior	Fall prevention
2	Self-care deficit: bathing/hygiene	Self-care: Activities of Daily Living (ADLs)	Self-care Assistance: bathing/hygiene
3	Impaired memory	Memory	Cognitive stimulation
4	Chronic confusion	Neurological status: consciousness	Medication management
5	Chronic confusion	Memory	Medication management
6	Impaired walking	Mobility	Exercise promotion: strength training
7	Impaired walking	Ambulation	Exercise promotion: strength training

## Data Availability

Not available.

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
