# Peer review of "Identifying Frequently Used NANDA-I Nursing Diagnoses, NOC Outcomes, NIC Interventions, and NNN Linkages for Nursing Home Residents in Korea"

_ijerph, 2021, doi:10.3390/ijerph182111505_

Round 1
Reviewer 1 Report
The objective of this paper was to identify and report the terminologies of NANDA-I diagnoses, NOC outcomes, NIC interventions, and NNN linkages that have been used for nursing homes residents with high frequency only for NH residents in Korea.
The study is based on survey completion and the results have been analyzed using the SPSS. The analysis is interesting and is complete from the methodological point of view.
I consider that the authors should stress in the paper (Conclusions) the importance of their findings and where and how these findings could be utilized.
I recommend the paper to be published.
Author Response
October, 23, 2021
Executive Editor, IJERPH
RE: Revision of Manuscript ID: ijerph-1411520 entitled " Identifying Frequently Used NANDA-I Nursing Diagnoses, NOC Outcomes,
NIC Interventions, and NNN Linkages for Nursing Home Residents in Korea"
To Whom It May Concern:
Attached please find an electronic copy of a second revision of Manuscript ID : ijerph-1411520 entitled, "Identifying Frequently Used NANDA-I Nursing Diagnoses, NOC Outcomes, NIC Interventions, and NNN Linkages for Nursing Home Residents in Korea," reviewed for publication in the IJERPH. In your editorial decision letter, you observed that the reviewers found considerable merit in the manuscript, but a number of issues required attention. You invited a revision and resubmission of the manuscript. We found your comments and those provided by the reviewers to be extremely helpful, and we hope you will find the revised manuscript to be substantially improved. Below, we provide a detailed point-by-point description of our response to each issue the reviewers raised.
I believe I have addressed each issue the reviewers raised and hope you find the revisions to be satisfactory. I look forward to hearing from you. Thank you for your time and consideration.
|
Reviewers' comments (Reviewer 1) |
Response to the comments |
Page and Lines |
|
I consider that the authors should stress in the paper (Conclusions) the importance of their findings and where and how these findings could be utilized. |
Thank you for your comment. We described the point. “The linkage can be applied to RNs’ independent activities with nursing languages to solve the potential or possible risk problems.” |
p.12 358-359 |

Reviewer 2 Report
Thank you very much for allowing me to review this manuscript. The research conducted is very interesting and highlights the NANDA, NIC and NOC diagnoses used in nursing homes by nurses.
As improvements to publish this work it is recommended
1. Indicate in the abstract data on the residents.
2. Explain how the NNN association was made. 3.
3. How was the selection of the resident and what did the selection procedure consist of?
4. To emphasize the practical implications in the conclusions.
Author Response
October, 18, 2021
Executive Editor, IJERPH
RE: Revision of Manuscript ID: ijerph-1411520 entitled " Identifying Frequently Used NANDA-I Nursing Diagnoses, NOC Outcomes,
NIC Interventions, and NNN Linkages for Nursing Home Residents in Korea"
To Whom It May Concern:
Attached please find an electronic copy of a second revision of Manuscript ID : ijerph-1411520 entitled, "Identifying Frequently Used NANDA-I Nursing Diagnoses, NOC Outcomes, NIC Interventions, and NNN Linkages for Nursing Home Residents in Korea," reviewed for publication in the IJERPH. In your editorial decision letter, you observed that the reviewers found considerable merit in the manuscript, but a number of issues required attention. You invited a revision and resubmission of the manuscript. We found your comments and those provided by the reviewers to be extremely helpful, and we hope you will find the revised manuscript to be substantially improved. Below, we provide a detailed point-by-point description of our response to each issue the reviewers raised.
I believe I have addressed each issue the reviewers raised and hope you find the revisions to be satisfactory. I look forward to hearing from you. Thank you for your time and consideration.
|
Indicate in the abstract data on the residents |
Thank you for your comment. We indicated the information of residents in the abstract. “Finally, the data of 57 residents in 25 NHs were collected.” |
p.1 14-15 |
|
Explain how the NNN association was made. |
Thank you for your comment. We described the point. “We integrated NNN association linkages by combining NIC (year) and NOC (year) taxonomies, in terms of the analogous domains and classes. We started with the NANDA-1 as an outline followed by the NIC and NOC. The whole process is parallel to the nursing process [25].” |
p.4 174-177 |
|
How was the selection of the resident and what did the selection procedure consist of? |
Thank you for your comment. We described the process of resident selection. “We randomly selected 140 NHs (20% of total NHs) using the random function of the Excel program and contacted NHs with RNs listed on the Korean Long-Term Care homepage operated by the Korean National Health Insurance Corporation [17].” “Each RN randomly selected one resident using the random-selection method. The criteria for including residents was those aged 65 or older who have been admitted for more than 6 months. We recruited RNs with more than 5 years of experience in NHs to participate.” |
p.2 87-89
p.3 97-100
|
|
To emphasize the practical implications in the conclusions. |
Thank you for your comment. We described the point. “The linkage can be applied to RNs’ independent activities with nursing languages to solve the potential or possible risk problems.” |
p.12 358-359 |

Reviewer 3 Report
Dear Authors,
Thank you very much for submitting the article for review. The work concerns the most commonly used tools for nursing diagnosis.
Unfortunately, the work is based on a very small group of respondents included in the survey.
The results presented are presented as percentages only. Only simple statistical analysis was used in the methodology.
the work is not innovative, it does not bring anything new in the knowledge of nursing and the entire health protection system.
Author Response
Please see the response in the attachment.

Reviewer 4 Report
The work carried out is very interesting because it shows the application of the specific professional language for nurses and with them the independent activity of the nurse for the diagnosis and potential or possible risk problems.
In the introduction, in my concept there are ideas that should be clarified or supported by specific bibliography that supports them, for example, when it is said that the nursing process is a problem-solving method. There is a similarity between the nursing process, the problem-solving method, and the scientific method, but each has application phases that have different conceptualizations and constructs. Each provides professional judgment: clinical, management or research. The authors should briefly provide the theoretical framework that supports their approach and that of the original Taxonomies and classifications, NANDA, NIC, NOC.
Methodologically if there are adaptations of the taxonomies to Korea. How they were carried out and how they were validated
When it is named that the observers received training or information and formation. How was inter-observer variability measured? It is important to describe what degree of prior consensus they had for the recovery of the writings in the patients' medical records.
It would be convenient to know which was the instrument in which the information was valued and its possible publication with the variables that were observed
Explain the random process in more detail
As an ethical concern, I am concerned that financial incentives have been offered to recruit participants. I may have misunderstood?
Author Response
October, 18, 2021
Executive Editor, IJERPH
RE: Revision of Manuscript ID: ijerph-1411520 entitled " Identifying Frequently Used NANDA-I Nursing Diagnoses, NOC Outcomes,
NIC Interventions, and NNN Linkages for Nursing Home Residents in Korea"
To Whom It May Concern:
Attached please find an electronic copy of a second revision of Manuscript ID : ijerph-1411520 entitled, "Identifying Frequently Used NANDA-I Nursing Diagnoses, NOC Outcomes, NIC Interventions, and NNN Linkages for Nursing Home Residents in Korea," reviewed for publication in the IJERPH. In your editorial decision letter, you observed that the reviewers found considerable merit in the manuscript, but a number of issues required attention. You invited a revision and resubmission of the manuscript. We found your comments and those provided by the reviewers to be extremely helpful, and we hope you will find the revised manuscript to be substantially improved. Below, we provide a detailed point-by-point description of our response to each issue the reviewers raised.
I believe I have addressed each issue the reviewers raised and hope you find the revisions to be satisfactory. I look forward to hearing from you. Thank you for your time and consideration.
|
The work is based on a very small group of respondents included in the survey. |
Thank you for your comment. We agreed your opinion and described this point in the limitation section. “Additionally, the number of subjects (RNs) and residents selected by the RNs included in this study is small to reflect whole NH residents. In future studies, it is necessary to secure a sufficient data-collection period and increase the number of nursing records of residents reviewed per RN.” |
p.10 348-341 |
|
The results presented are presented as percentages only. Only simple statistical analysis was used in the methodology. the work is not innovative, it does not bring anything new in the knowledge of nursing and the entire health protection system. |
I totally agree with your opinion. I hope you understand our results based on our explanation.
We integrated NNN association linkages by combining NIC (year) and NOC (year) taxonomies, in terms of the analogous domains and classes. We started with the NANDA-1 as an outline followed by the NIC and NOC. The whole process is parallel to the nursing process. Thus, it can be presented only as percentages. |
- |
|
The work carried out is very interesting because it shows the application of the specific professional language for nurses and with them the independent activity of the nurse for the diagnosis and potential or possible risk problems. |
I agree with you. I explained this in the discussion section. “The linkage can be applied to RNs’ independent activities with nursing languages to solve the potential or possible risk problems” |
P 10 358-359
|
|
In the introduction, in my concept there are ideas that should be clarified or supported by specific bibliography that supports them, for example, when it is said that the nursing process is a problem-solving method. There is a similarity between the nursing process, the problem-solving method, and the scientific method, but each has application phases that have different conceptualizations and constructs. Each provides professional judgment: clinical, management or research. |
Thank you for your comment. The unclear sentences were revised. “The nursing process is a systematic method that allows RNs to effectively care for patients using problem solving and critical thinking [2]. The aim of the nursing process is identifying, diagnosing, and treating actual or potential human responses to disease [3]. The nursing process helps nurses make professional judgments in terms of clinical and problem-solving methods and nursing management [2,3].” |
p.1 28-33 |
|
The authors should briefly provide the theoretical framework that supports their approach and that of the original Taxonomies and classifications, NANDA, NIC, NOC. |
Thank you for your comment. We described the point. “The conceptual framework of this study is the Structure-Process-Outcome (SPO) model developed by Unruh and Wan [16] for evaluating the quality of long-term care facilities. NANDA is classified in the structure, NIC is equivalent to the process, and NOC belongs to the outcome.” |
p.2 82-85 |
|
Methodologically if there are adaptations of the taxonomies to Korea. How they were carried out and how they were validated. |
Thank you for your comment. In Korea, studies have been conducted on what nursing diagnoses, interventions, and outcomes are used and how they align in various settings such as military hospitals, surgical nursing units, home healthcare settings, and obstetric department nursing units, but they are not actually applied in clinical practice. Continuously, research should be conducted and verified and applied to clinical practice. |
- |
|
When it is named that the observers received training or information and formation. How was inter-observer variability measured? It is important to describe what degree of prior consensus they had for the recovery of the writings in the patients' medical records. |
Thank you for your comment. We described the process of subject (RN) training and inter-observer variability. “We explained the definition, purpose, and brief history of nursing diagnoses, outcomes, and interventions using a booklet and how we developed questionnaires before providing the survey so RNs had no difficulty filling out the questionnaire..” “For the interrater reliability of the data, two RNs independently checked nursing diagnoses, outcomes, and interventions on the questionnaire. If they chose different nursing diagnoses, outcomes, and interventions, they were unified into one after the two RNs discussed their findings. This process secured the reliability between the observers.” |
p. 4 156-159
p. 4 164-168
|
|
It would be convenient to know which was the instrument in which the information was valued and its possible publication with the variables that were observed |
Thank you for your comment. In this study, simply a list of nursing diagnoses, interventions, and outcomes was presented and selected among the RNs. This is the instrument. The presented list selected by the RNs is presented in Tables 2 and 3. |
|
|
Explain the random process in more detail |
Thank you for your comment. We described trandom process in more detail. “We randomly selected 140 NHs (20% of total NHs) using the random function of the Excel program and contacted NHs with RNs listed on the Korean Long-Term Care homepage operated by the Korean National Health Insurance Corporation [17].” “Each RN randomly selected one resident using the random-selection method. The criteria for including residents was those aged 65 or older who have been admitted for more than 6 months. We recruited RNs with more than 5 years of experience in NHs to participate.” |
p.2 87-89
p.3 97-100
|
|
As an ethical concern, I am concerned that financial incentives have been offered to recruit participants. I may have misunderstood? |
Thank you for your thoughtful comment. It is not used to recruit the participants. We mentioned that we would offer financial incentives, but the incentive was provided only after they completed the surveys. |
p.3 101-102 |

Round 2
Reviewer 2 Report
The corrections indicated by the reviewers have been made. Therefore, the manuscript should be accepted.